# Effect of Ca Precipitation on Texture Component Development in AZ Magnesium Alloy

**DOI:** 10.3390/ma15155367

**Published:** 2022-08-04

**Authors:** Kibeom Kim, Yebin Ji, Kwonhoo Kim

**Affiliations:** 1Department of Marine Design Convergence Engineering, Pukyong National University, 45 Yongso-ro, Nam-gu, Busan 48513, Korea; 2Department of Metallurgical Engineering, Pukyong National University, 45 Yongso-ro, Nam-gu, Busan 48513, Korea

**Keywords:** Al_2_Ca, Ca precipitation, dynamic recrystallization, EBSD, high temperature deformation, particle stimulated nucleation, crystallographic texture

## Abstract

To enhance the formability of magnesium alloys, inhibition of basal texture development by the particle-stimulated nucleation (PSN) effect has attracted significant interest. However, its contribution to texture development is not easily observed due to the separation of texture from the conventional deformation behavior. This study aims to separate the Ca texture from the deformation behavior of AZX611 alloy and quantify it using scanning electron microscopy with electron backscatter diffraction (SEM-EBSD). Since Ca in the AZ61 magnesium alloy precipitated as Al_2_Ca, the hot-rolled magnesium alloys AZ31, AZ61, and AZX611 were used. High temperature compression was conducted at 723 K, the strain rate 0.05/s and 0.005/s and the true strain up to −1.0. Dynamic recrystallization was observed in each specimen and the Ca-free alloys showed dislocation glide at high strain rates and solute drag at low strain rates. When the dislocation glide dominated, basal texture was strengthened. In contrast, solute drag caused non-basal texture development. Precipitation hardening caused AZ61 to have higher flow stress than those of the Ca-free alloys by the PSN effect; its texture was observed separately because the PSN grain growth around the precipitation and orientation was specific, similar to the one developed at the solute atom drag.

## 1. Introduction

Although magnesium alloys have a high specific strength and low density, an insufficient slip system of hexagonal close packed (HCP) structure has limited their application as a structural material [1,2,3]. Moreover, basal texture of magnesium formed after high temperature process is a major cause for plastic anisotropy and low formability such as fracture or cracking. Therefore, it is essential to understand and control the mechanism of crystallographic texture formation [4,5]. Especially, H. Fukutomi et al. studied the relationship between deformation condition and texture formation mechanism using various type of crystal structure such as Al for face-centered cubic, Mg for HCP crystals and proposed the “Dynamic Prior Grain Growth (DPGG)” mechanism [6,7,8,9,10].

It was found that factors such as shearing stress differences, crystal orientation, activated slip systems, and stacking fault energies can affect the stable orientation which can be the major texture component after deformation. It was suggested that there must be a relationship between the strain rate and the mechanism of dislocation movement during the creep mechanism [11,12]. When the strain rate is high or the temperature is low, the stress exponent (*n*) became almost 5 or more and the dislocation glide became the major mechanism. In this case, the basal texture was set as a stable orientation and strengthened during the high temperature deformation. In contrast, when the *n* value neared 3, solute atom drag became the major mechanism and different texture component was developed.

This seemed to have been governed by deformation rather than recrystallization. Although the ideal non-basal texture can be developed by changing the *n* value, the application of these conditions is difficult owing to the low deformation rate productivity.

On the other hand, many studies have been conducted to change the basal texture during deformation. Among them, element alloying was of high interest because of process convenience and ability to assign various properties. Especially, rare-earth (RE) elements have been reported as a modifier of the basal texture to RE-texture [13,14]. Similarly, Ca has been reported for the inhibition of basal texture development by the effect of precipitation, so called particle stimulated nucleation (PSN), instead of solid solution as well as the RE elements [15,16]. However, conventional methods had difficulty to observe the PSN texture directly, because the extremely fine grains are formed, and their size becomes similar with the grains formed conventionally in the end. The fine grains are hard to be observed independently by the X-ray diffraction, which is a method for macro scale texture. Similarly, although EBSD method can observe the grain orientation of each point under the specific condition such as grain size, lowered step size and map scale are hard to show the texture. Therefore, it seemed that other method needs to be adapted.

In our previous study, through the high temperature annealing in the 773 K, the Ca-based precipitate was spheroidized and distributed to the overall matrix with lowered size. Moreover, magnesium matrix had a higher grain size than the conventional annealing condition [17,18]. Therefore, it was expected that the PSN texture could be separated from that of the whole deformation behavior using this condition.

Thus in this study, using field-emission scanning electron microscope (FE-SEM) and electron back scattered diffraction (EBSD), PSN texture was separated and quantified from the deformation behavior of AZX611 during high temperature deformation.

## 2. Materials and Methods

The commercial cast AZ31 (Mg-3.0 wt.% Al-1.0 wt.% Zn), AZ61 (Mg-6.0 wt.% Al-1.0 wt.% Zn), and AZX611 (Mg-6.0 wt.% Al-1.0 wt.% Zn-1.5 wt.% Ca) were used as the starting material. The investigation about the effect of Ca element on phase composition was conducted prior to plane strain compression. A Mg-Ca phase diagram for a state with approximately 6 wt.% Al was obtained using FactSage (ver. 8.0, CRCT, Quebec, Canada), a phase diagram analysis software.

All alloys were hot rolled at 673 K and almost 30% rolling reduction. After rolling, the alloys were machined to rectangular shape withe 6.7 × 10 × 10 mm. The specimens were sealed in a quartz tube with Ar gas and annealed at 773 K-4 H for stress relaxation and second-phase distribution. After then, they were quenched in water. Most grains of the alloys were fully annealed and equiaxed. To investigate the Ca distribution in the deformation specimen, As-received specimen of AZX611 was observed using field-emission electron probe micro analyzer (JXA-8530F, JEOL, Tokyo, Japan).

High-temperature plane strain compression was conducted with the universal testing machine and channel die. The compression direction of specimen was along the normal direction of the rolling plate and the elongation of transverse direction was restricted using the channel die. The compression was conducted at 723 K and strain rates of 5 × 10^−2^ s^−1^ and 5 × 10^−3^ s^−1^. The stress was applied to the specimen until the true strain became −0.4, −0.7 and −1.0 each. The restricted plane (the transverse direction (TD) plane of the rolled plate) of each deformed specimens was cut to observe the microstructure, and the compression plane (ND plane of rolled plate) was cut for texture observation.

They were mechanically grinded using conventional method and electrochemically polished in perchloric acid solution. For electropolishing, a solution of 2~4% concentration of perchloric acid with four-nine purity ethanol was used, and the current density was set to 0.18–0.22 A/cm^2^ by controlling the voltage at 243 K. The microstructure and texture were observed using FE-SEM (CLARA, TESCAN, Brno, Czech Republic) and EBSD (Symmetry S2, Oxford instrument, Abingdon, OX, UK), respectively. The texture analysis using EBSD was conducted under the 8 μm step size to the square area with sides of length 1600 μm of ND plane. About 40,000 points were investigated in each image and only datasets over 90% initial indexing rate were used. In addition, any filter or cleaning was not adopted.

Figure 1 shows the inverse pole figure and microstructure of specimens before deformation. The main texture components are shown on an (α, β) coordinate system, where α signifies the angle between [0001] and [10$\bar{1}$0] and β signifies the angle between [10$\bar{1}$0] and [11$\bar{2}$0]. Therefore, the main texture components of all initiative specimen were basal and (0, 0). Furthermore, the second phase of AZX611 decomposed as small particles under 4 μm.

## 3. Results

### 3.1. Phase Composition in Ca Alloy

Figure 2 shows the Mg-Ca phase diagram with 6 wt.% Al, which is obtained using Fact Sage 8.0. About 1.5 wt.% Ca concentration was used in the AZX611 alloy, but the Ca solution zone is absent in the phase diagram. In the cooling or heating path of this composition, the secondary phases Mg_17_Al_12_ and C15 (Al_2_Ca) could be formed.

According to the phase diagram, Mg_17_Al_12_ and Al_2_Ca forms or decomposes around 543 K and 783 K, respectively. Therefore, AZX611 is composed of two different phases, Mg matrix and Al_2_Ca precipitates, at 723 K where the compression was conducted. In addition, because there is no solid solution zone of Ca in the matrix, the Al content was recalculated for each phase. As a result, approximately 4 wt.% Al was in the matrix while 2 wt.% Al was in the precipitates.

Figure 3 is the result of measuring specific area of the microstructure shown in Figure 1d using EPMA. In Figure 3a, it was confirmed that the precipitates were decomposed by annealing. In addition, Figure 3b–d shows the distribution of Mg, Al, and Ca, which are the main elements of AZX611 alloy. At the point where the precipitates occurred, the peak of Mg was very low compared to other matrix, and Al appeared in both the precipitates and the substrate, but showed very high strength especially in the part where the precipitates were formed. In addition, in the case of Ca, the peak was hardly confirmed in the substrate, and it could be confirmed that it appeared enough to be observed only in the portion with precipitates. This phenomenon well supported that Ca only contributed to the formation of precipitates and that there was no solid solution region in this composition.

### 3.2. True Stress-Strain Curve

Figure 4 shows the true stress-stain curves for the plane strain compression. True strains are shown as absolute values. In the high strain rate deformation, all flow stresses in the high strain rate showed intense increment during initial strain and work softening after peak stress while an obvious peak stress was not observed in the low strain rate. AZX611 had the highest flow stress whereas those of AZ31 and AZ61 were nearly identical, and although AZ61 has a higher Al concentration than that of AZ31, its flow stress was lower.

At the lower strain rate, AZ61 had stress values lower than 20 MPa. Moreover, AZX611 had higher stress values than those of the Ca-free alloys AZ31 and AZ61, even though it has the second highest Al concentration of the three, this mean that Al_2_Ca contributed to precipitation hardening.

### 3.3. Microstructure Development

Figure 5 shows the microstructure of the TD plane of AZ61 and AZX611 after deformation at different strain rate condition. Each condition of strain rate and alloy is indicated above each figure. Each grain orientation was colored using the standard triangle and shows the ND direction. Grain misorientation over 15° was defined as high angle boundary (HAGB) and shown with thick black lines, while the range of 5° to 15° was considered a low angle grain boundary (LAGB) and shown with grey lines. Similar results have been reported under the same deformation condition in other AZ series magnesium alloys or Al-Mg alloys [9,19], thus, this deformation condition seems to undergo dynamic recrystallization (DRX). All grains were refined by the DRX during the compression, they were observed especially high in AZX611. More specific DRX behavior will be discussed in the later section.

Figure 6 shows the kernel average misorientation (KAM) distribution of the microstructure shown in Figure 5a,b. The KAM maps describe the average misorientation between each pixel and gives a high misorientation value in areas where deformation is in progress, such as at grain boundary migrations [19]. Misorientations range from 0° to 5° are shown by color scale.

In Figure 6a, the high densities of misorientation were mainly concentrated on the grain boundaries either around the LAGB or HAGB. Figure 6b is a KAM map of the low strain rate condition and its behavior was different from that of the dislocation glide. In contrast with the large difference in misorientation occurred around the grain boundary where the dislocation glide was dominant, it was high both at the grain boundary and within the grain boundary. Moreover, the formation of new grain boundary and the growth of them were less shown.

Figure 6c is the microstructure of AZ61 in the earlier stage of Figure 5a. Grains were more in coarsen. In addition, Figure 6d shows the texture of the specimen in Figure 5b. The texture of the non-basal texture was formed as (22, 0) at this time, whereas the component of the texture was maintained as (0, 0) which was initially formed in high temperature rolling.

Figure 7a,b shows the band contrast and KAM distribution, respectively, of the microstructure shown in Figure 5c. In the band contrast image, the distribution of precipitation can be seen, with precipitations being especially concentrated in the dark box. In addition, misorientation appeared mostly around the grain boundaries and the precipitation area in the box as shown in the KAM distribution.

Figure 8 shows the grain size distribution of AZ61 and AZX611 at strains of 0.4 and 0.7. During the process of DRX in AZ61, grain size was lowered while maintaining a normal distribution. In comparison, because the DRX grains grew up from the lower grain sizes in AZX611, the fraction of grains of 16 μm or less increased, while that of grains larger than 16 μm decreased.

### 3.4. Texture Development

Figure 9 shows the texture intensity and main component of each specimen deformed at high strain rate. Texture component is expressed below each point using an (α, β) coordinate system. Both AZX611 and Ca-free alloys maintained their main texture component as (0, 0) during the overall strain. However, their texture intensity varied differently. Intensities of Ca-free alloy were increased continuously along with the strain increase. Furthermore, the texture intensity of AZ61 increased more sharply than that of AZ31. AZX611 increased with strain up to −0.4 but decreased gradually at higher strain values.

## 4. Discussion

### 4.1. Deformation Mechansim

As assumed in Section 3.1, because Ca did not solute in to AZX611, the major alloying element was Al in Mg matrix. Therefore, matrix of AZX611 would have same behavior with Ca-free alloy. To discuss the effect of Ca precipitation of AZX611′s deformation behavior, it was needed to separate the effect of Al content in matrix.

Dislocation and twinning operate commonly as mechanism which accepts the deformation in Mg alloy. Especially, twinning allows for further basal slip or recrystallization. However, at the temperature over 708 K, CRSS for non-basal slip becomes lower than twinning and it becomes hard to work [20]. In this study, it was hard to observe the twinning in every specimen and thus dislocation was regarded as the major deformation mechanism. Through the as-received specimen shown in Figure 1a–c, it was confirmed that all of specimens had (0, 0) basal texture which is parallel to compression direction. During the deformation, <a> slip systems including basal, prismatic were hard to be activated due to low CRSS from geometrical condition. This mean that <c + a> slip system became dominant.

### 4.2. Deformation Behavior of Mg-Al Matrix

In the true stress-strain curves of AZ61 compressed with different strain rates, their stress was different along with the strain rate. Different texture component was developed after the true strain up to −1.0. This means that strain rate can affect the mechanism of dislocation movement. It has been reported and discussed that the creep mechanism has high relationship with this phenomenon in AZ80 [10].

When the dislocation glide dominates the deformation, dislocation is able to be inhibited by the grain boundaries and piles up around them. They lead to misorientation accumulation and induce subgrain rotation and grain boundary migration. Eventually, HAGBs will be developed from LAGBs and continuous DRX will occur. Similarly, in Figure 6a, it can be seen that such a misorientation was mainly observed around the grain boundaries. These phenomena confirmed that the dislocation moved via the glide model and that continuous recrystallization occurred.

On the contrary, peak stress decreased under 20 MPa and the non-basal texture component was developed when the strain rate decreased. This phenomenon was considered a conversion of dislocation mechanism to solute drag as the strain rate decreased. When the solute atom drag dominates the deformation behavior as the strain rate decreases, solute atoms react with the dislocation by relaxing the stress field. This eliminates the driving force, enabling the formation of new grain boundaries from conventional ones. This implies that the non-basal texture might be developed by the deformation rather than recrystallization. During the microstructure change by DRX under the dislocation glide mechanism, the basal textures maintained their components and strengthened during the deformation process. It was implied that in this condition, grains with the [0001] orientation were set as the stable orientation and grain growth was directed toward this orientation.

To be set as the stable orientation, two independent conditions must be fulfilled: (1) arrangement of the grains to have a low Taylor factor which do not change during the deformation, and (2) every slip system activated in the same family during deformation should equilibrate under stress to prevent crystal rotation.

Although CRSS values become almost similar at 723 K, the Schmidt factor and its stress are applied highly on <c + a> slip systems because of the geometrical structure of HCP. This causes the grains with orientations near (0001) to have a lower Taylor factor than that of others. Furthermore, grains with low Taylor factor are regarded as having low values of stored energy for deformation and are less sensitive to shearing. Thus, basal grains were able to accept the deformation only through grain boundary migration and growth consuming the grains with high values of stored energy.

In contrast, even if the number of activated slip systems satisfies the von Mises criteria and can accommodate the deformation, if one of the slip systems is receiving higher stress than that of its family system, the crystal rotation would shift toward its direction [21]. Results of AZ31 plane strain compression have shown that when their activated slip systems set as the prismatic <a>, but each slip system is in a non-equilibrium state, their texture component rotates during the strain is becoming increased [6,10].

However, in this case, texture development was continued until the final strain, and there was no evidence that crystal rotation occurred. Eventually, basal texture was maintained and increased because of these reasons.

### 4.3. Effect of Ca Precipitation on Deformation Behavior

Until the previous section, it was concluded that: (1) Ca did not solute into the matrix, but contributed to the formation of the Al_2_Ca phase, and the matrix of AZX611, with the second highest Al concentration compared to those of AZ31 and AZ61, showed the same behavior; (2) dislocation glide was the major deformation mechanism with the strain rate 5 × 10^−2^ s^−1^ and the deformation was accomplished by the <c + a> slip system and continuous DRX; (3) microstructure and texture deformation was altered by the Al_2_Ca. In this section, the effect of the Al_2_Ca phase on deformation behavior will be discussed based on these results.

As shown in Figure 5c, DRX occurred in all alloys and the grains in AZX611 were especially refined. Although continuous DRX was observed in the matrix, many areas that have particularly concentrated fine grains were also observed. In the true stress-strain curve, the Al_2_Ca phase caused AZX611 to have higher flow stress than that those of the Ca-free alloys by precipitation hardening, and this mean precipitation can act as an obstacle to dislocation.

In the previous section, it was found that continuous DRX occurred by subgrain rotation and grain boundary migration because of the grain boundary inhibiting the glide of dislocation. Similarly, the area around the precipitation can also act similar to a boundary. The misorientation would pile up to the further surface and provide the new sites and driving force for nucleation. Different types of DRX would then progress. In this study, newly formed grains developed just around the Ca precipitation, whereas in conventional DRX in the single phase, they were formed both at the grain boundaries and ingrain. The new type of DRX that occurred in this case is known as the particle stimulated nucleation (PSN) effect [16,22].

In the development of texture in AZX611 as shown at Figure 7, the matrix of AZX611 underwent the same deformation behavior as did the Ca-free alloys; however, its intensities varied. It was same that intensity increased in initial strain stage and the increment of (0, 0) intensity happened from the continuous DRX, but at strain values greater than −0.4, the intensity weakened. This seems to be the reason that the new grains formed by PSN have different orientation from the conventional [0001] axis causing the axis density to weaken. It was confirmed that the recrystallization behavior of continuous DRX and PSN recrystallization was different. To separate the texture component caused by the PSN effect from whole microstructure, the grain distribution was analyzed by measuring the ND plane of specimen.

Figure 10 shows the microstructures and inverse pole figures (IPF) of grains under and over 16 μm of the deformed AZX611 specimen with strain at −0.7. Each IPF has a scale, with the scale bar placed above them. Grains over 16 μm had a texture of (0, 0) and a higher intensity than that of grains under 16 μm, which had a texture of (20, 0–30).

The relationship between texture intensities and true strain is shown in the graph in Figure 11. Textures are depicted as 3 types overall and divided at the grain size 16 μm. True strain is set from −0.7 to −1.0 and −0.1 point each. Texture component is indicated at each point using the coordinate system. Fractions of fine grains are expressed below each point. After strain −0.7, the overall texture component was (0, 0) and intensity decreased continuously. Coarse grains over 16 μm showed stronger basal texture than that of the overall texture. However, the intensity of grains under 16 μm, which have a (20, 0–30) component, became stronger and the fractions increased. These series of phenomenon suggest that the PSN effect formed another orientation, and the development of basal texture was inhibited by this.

Although the dislocation glide mechanism dominates, the newly formed grains of the PSN effect had an aligned texture to a specific orientation that was not random. This mean that Ca precipitation changed the conventional [0001] stable orientation. The texture component of AZ61 at the low strain rate was demonstrated to be (22, 0) and Helis et al. had reported that (19, 0) and (21, 30) texture occurred in AZ31 at the same strain rate and initial texture condition [6]. Thus, in this study, the (20, 0–30) texture found in AZX611 was seemed to be a mixture of the two components and the precipitation rolled similar to that of the solute drag mechanism.

The PSN texture was separated from the overall deformation using different initial conditions and recrystallization behaviors. Due to the relatively large step size among the measurement conditions, it is possible that the separated PSN texture data still includes the one occurred from cDRX in meantime. Nevertheless, even in the high step size condition, the non-basal orientation can be observed without consumption by the stable orientation [0001] during the deformation.

In the solute drag mechanism, the component of deformed texture in magnesium alloy is highly related to the activation of slip systems interacting with the solute atoms. This means that a combination slip system satisfying the Von Mises criteria changes, so that the Taylor factor—which shows the stable orientation—would be changed. However, which change to the slip system would occur still remains as a problem.

To analyze the verification of the slip system combination and calculate the Taylor factor among these deformations, it could be further specified by performing compression according to the orientation conditions of the initial basal texture. Although Al_2_Ca was discomposed and dispersed by annealing in the as-received specimens, the PSN effect occurred in specific areas. Some areas where grain refinement by the PSN effect did not appear were observed in same time. Due to the structure differences between the Mg matrix (HCP) and Al_2_Ca (Cubic), the lattice relationship between both phases differs from the orientation relationship. Therefore, it would be simpler to specify the change in Taylor factor if the observation of the effect of initial texture using this alloy can be conducted.

## 5. Conclusions

In conclusion, the texture component developed from Ca precipitation during plane strain compression of AZ61 magnesium alloy was analyzed using SEM-EBSD and the texture caused by the PSN effect was successfully studied separately from that of the deformation behaviors. The major results are as follows:The Ca in AZX611 only contributed to the formation of Al_2_Ca precipitate at the temperature 723 K, lowering the Al concentration in the matrix. Due to the Al_2_Ca, AZX611 showed higher flow stress than that of AZ61.Dynamic recrystallization occurred in every alloy during the deformation, Al_2_Ca induced the PSN effect, and because of the coarse grain size of the initial state, different types of recrystallization textures were separated from each other.The dislocation glide mechanism dominated the deformation behavior in the high strain rate condition in this study and the basal texture strengthened. However, in AZX611, newly recrystallized grains formed by the PSN effect had a different texture that can weaken the basal texture.The non-basal component of PSN texture occurred during the dislocation glide mechanism, and it was similar with the texture component formed in the solute atom drag mechanism.

## Figures and Tables

**Figure 1 materials-15-05367-f001:**
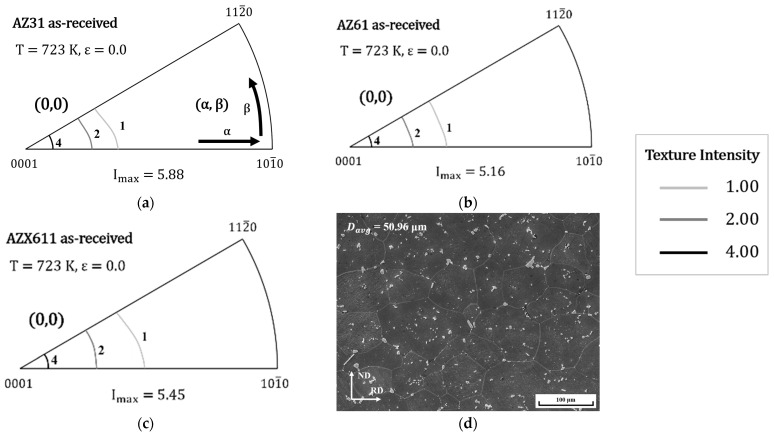
The (0001) pole figure of (**a**) AZ31, (**b**) AZ61 and (**c**) AZX611 as-received specimen and (**d**) the microstructure of AZX611.

**Figure 2 materials-15-05367-f002:**
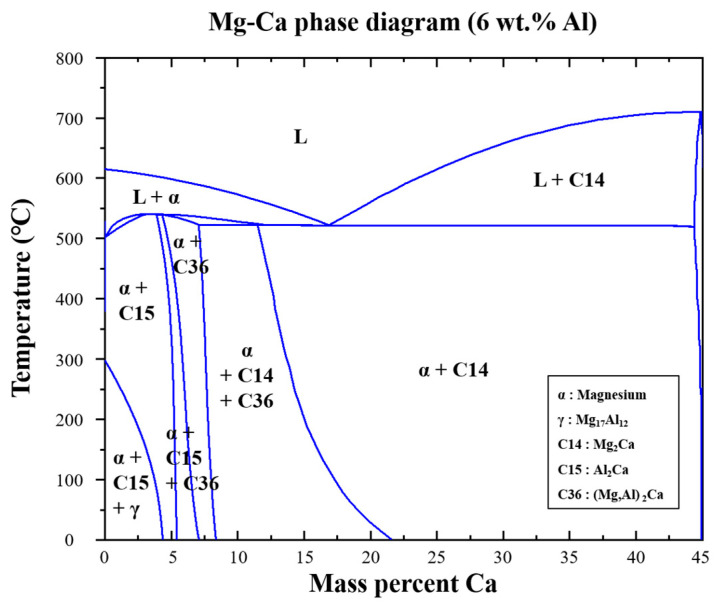
Phase diagram of Mg-Ca binary system at 6 wt.% Al content.

**Figure 3 materials-15-05367-f003:**
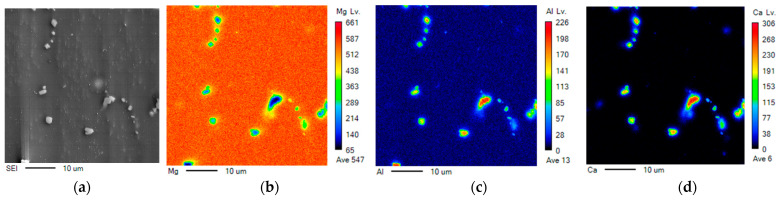
The result of EPMA in AZX611 as-received specimen shown in Figure 1d. Each figure shows the (**a**) SEI image and the distribution of (**b**) Mg, (**c**) Al and (**d**) Ca element.

**Figure 4 materials-15-05367-f004:**
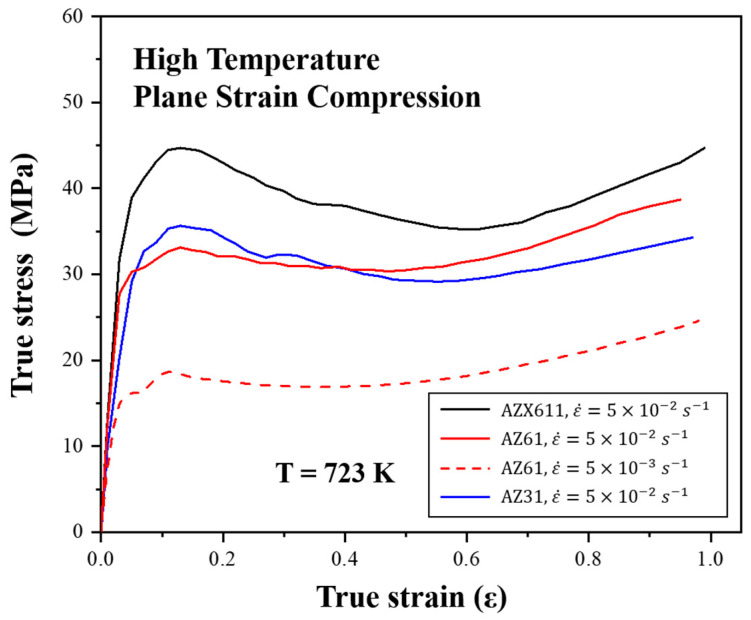
True stress-strain curves for the deformation at 723 K up to the strain of −1.0 with each strain rate.

**Figure 5 materials-15-05367-f005:**
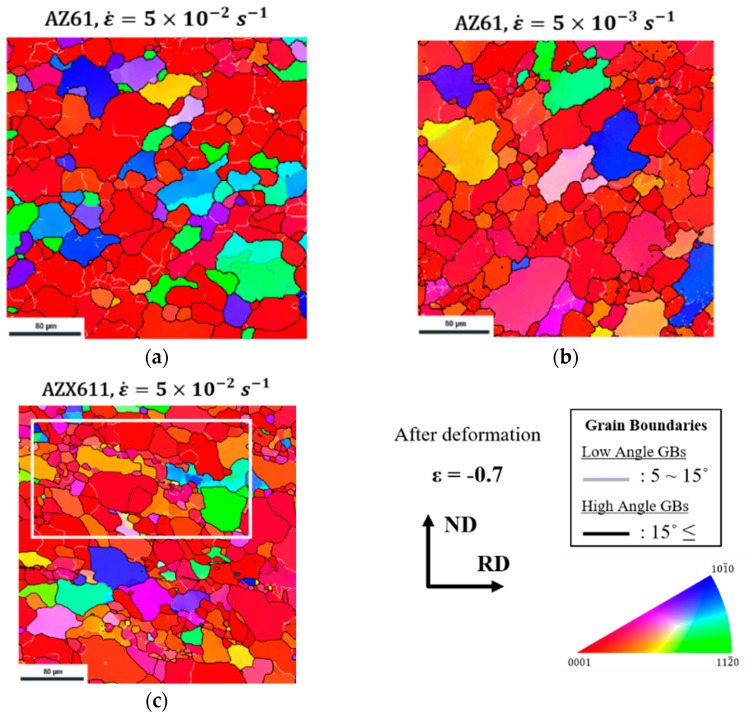
Microstructures of the TD planes of (**a**), (**b**) AZ61 and (**c**) AZX611 specimen TD plane after deformation strain up to −0.7 in each strain rate.

**Figure 6 materials-15-05367-f006:**
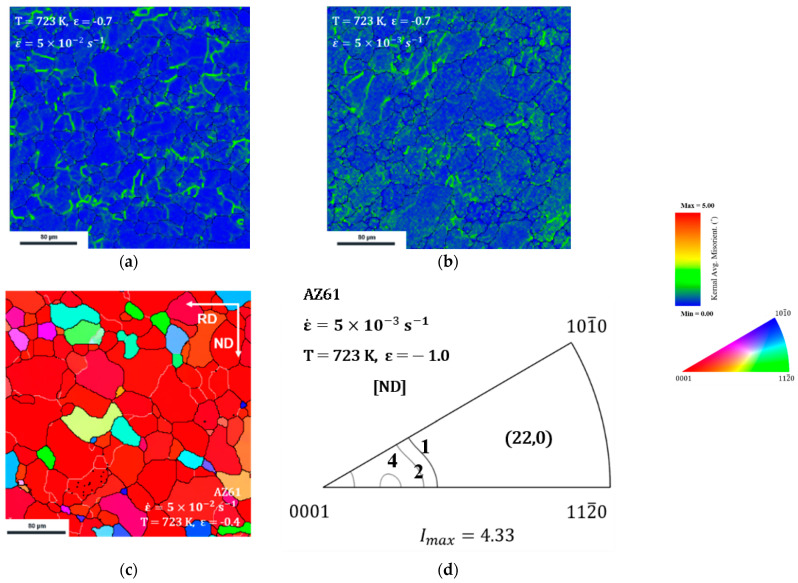
The KAM map of AZ61 specimen after deformation at the (**a**) high and (**b**) low strain rate, (**c**) the microstructure of AZ61 after deformation strain up to −0.4 and (**d**) texture of specimen after deformation in the low strain.

**Figure 7 materials-15-05367-f007:**
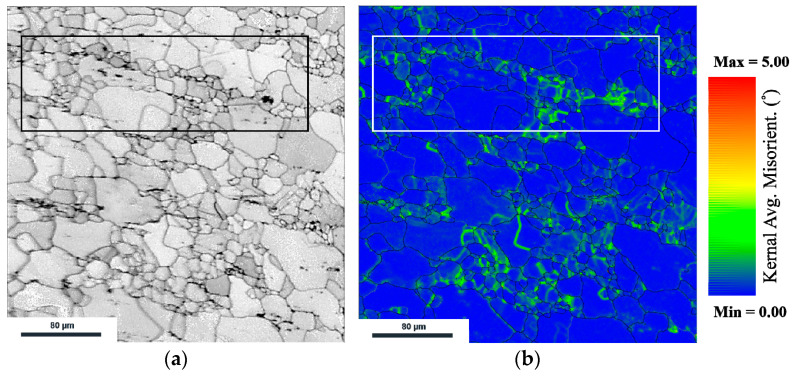
The (**a**) band contrast image and (**b**) KAM map of AZX611 specimen shown in Figure 5c.

**Figure 8 materials-15-05367-f008:**
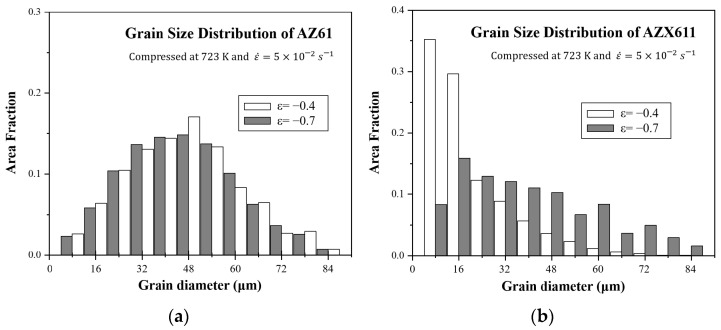
The graphs showing grain size distribution of (**a**) AZ61 and (**b**) AZX611 after each deformation condition.

**Figure 9 materials-15-05367-f009:**
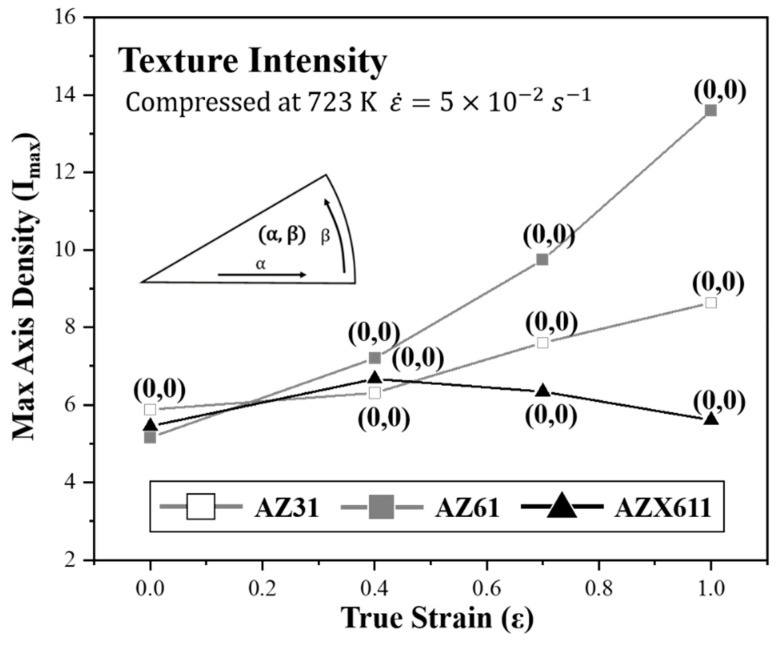
The graph showing the relationship between texture intensity and each strain. All specimens were deformed at high strain rate. Main texture component is shown below each point.

**Figure 10 materials-15-05367-f010:**
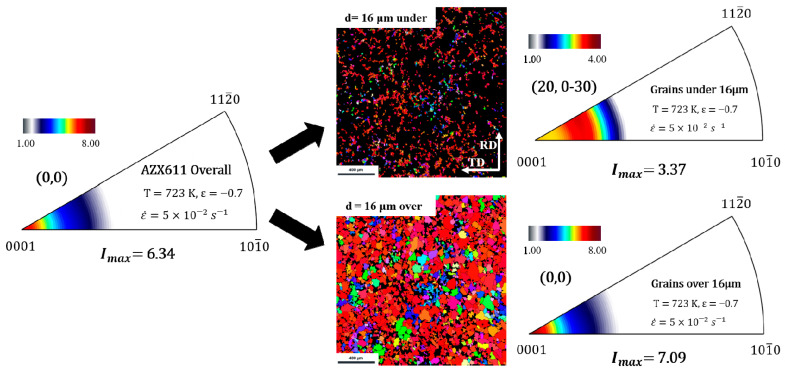
IPF of AZX611 deformed specimens after the strain −0.7. Each IPF shows the texture of grains overall and separated along with the grain size.

**Figure 11 materials-15-05367-f011:**
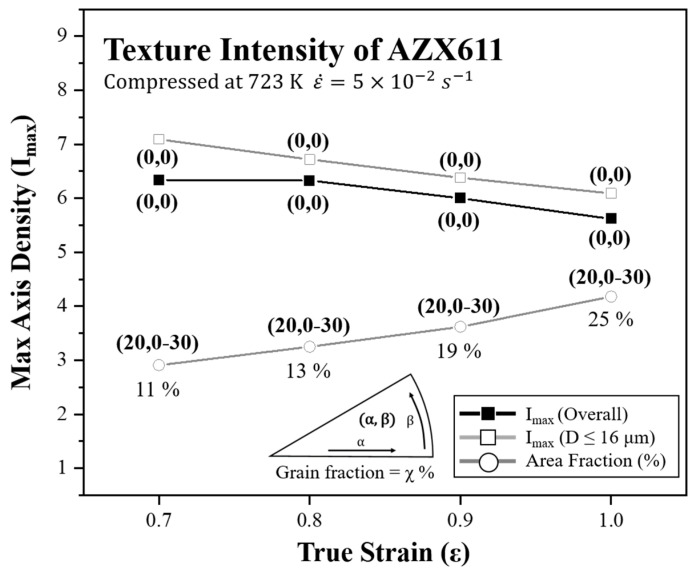
The graph showing fraction and texture intensity of grains under 16 μm.

## Data Availability

The datasets generated during and/or analyzed during the current study are available from the corresponding author on reasonable request.

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
