# Peer review of "Effect of Ca Precipitation on Texture Component Development in AZ Magnesium Alloy"

_materials, 2022, doi:10.3390/ma15155367_

Round 1

Reviewer 1 Report

In the manuscript, the authors have attempted to separate the Ca texture from the deformation behavior of AZX611 alloy and quantify it using scanning electron microscopy with electron backscatter diffraction.

However, a few modifications below can further enhance the quality of the manuscript. 

What equipments for FE-SEM and EBSD was used?

EDX mapping of all samples should be indicated.

Grammar check, spell check and punctuation also needs to be checked thorough out the entire manuscript

Author Response

We have reviewed the comments provided, and we have corrected all points. Response details to reviewers' comments were written in "Response Letter". Please see the attachment.

Reviewer 2 Report

This research paper was aimed at studying the effect of Effect of Ca precipitation on texture component development in AZ magnesium alloy. The authors did a very comprehensive and well-articulated study which I endorse its publication. However, minor corrections are recommended as follow:

1. Ln 70 – 71: Recast this sentence “Because precipitation was observed in the AZX611 cast alloy ....”

2. Ln 62 - 63: “...it 62 was objected to separate the Ca texture from the ...” objected cannot be used as “aimed”. So, re-cast the sentence.

3. The gap in knowledge of this research needs to be projected more vividly in the introduction.

4. It was stated in the work that “The PSN texture seemed to be formed by the deformation rather than re-crystallization”. Can you back this observation up with other researcher’s views? Reference it appropriately.

Author Response

(The authors gave the same response as above.)
